# An Evolutionary Conservation and Druggability Analysis of Enzymes Belonging to the Bacterial Shikimate Pathway

**DOI:** 10.3390/antibiotics11050675

**Published:** 2022-05-17

**Authors:** Rok Frlan

**Affiliations:** The Department of Pharmaceutical Chemistry, Faculty of Pharmacy, University of Ljubljana, 1000 Ljubljana, Slovenia; rok.frlan@ffa.uni-lj.si; Tel.: +386-1-4769-674

**Keywords:** shikimate pathway, druggability, ligandability, antibacterial, inhibitors, broad-spectrum

## Abstract

Enzymes belonging to the shikimate pathway have long been considered promising targets for antibacterial drugs because they have no counterpart in mammals and are essential for bacterial growth and virulence. However, despite decades of research, there are currently no clinically relevant antibacterial drugs targeting any of these enzymes, and there are legitimate concerns about whether they are sufficiently druggable, i.e., whether they can be adequately modulated by small and potent drug-like molecules. In the present work, in silico analyses combining evolutionary conservation and druggability are performed to determine whether these enzymes are candidates for broad-spectrum antibacterial therapy. The results presented here indicate that the substrate-binding sites of most enzymes in this pathway are suitable drug targets because of their reasonable conservation and druggability scores. An exception was the substrate-binding site of 3-deoxy-*D*-arabino-heptulosonate-7-phosphate synthase, which was found to be undruggable because of its high content of charged residues and extremely high overall polarity. Although the presented study was designed from the perspective of broad-spectrum antibacterial drug development, this workflow can be readily applied to any antimicrobial target analysis, whether narrow- or broad-spectrum. Moreover, this research also contributes to a deeper understanding of these enzymes and provides valuable insights into their properties.

## 1. Introduction

Although antibiotics have saved many lives in the past, their overuse and misuse have led to an increased economic burden as well as increases in morbidity and mortality in patients infected with formerly common infections that are not treatable with existing treatment options [1]. There are many promising strategies in the fight against bacterial pathogens that have evolved over the past decades, but the one that is most frequently mentioned in the scientific literature is targeting an essential metabolic pathway that is not exploited by existing drugs [2]. In this context, the enzymes of the shikimate pathway may be a possible solution because they have no counterpart in mammals and are essential for growth and virulence, as shown by several knock-out studies [3,4,5,6,7,8,9]. Importantly, this pathway was discovered more than 70 years ago and has been studied in detail since then [10,11,12,13,14].

An overview of literature data suggests that the interest in these enzymes as antibacterial targets has increased over the years (Appendix A). However, these enzymes have been largely ignored by the pharmaceutical industry, which to date has been unable to produce a clinically relevant drug against any of them [15]. Although many inhibitors have been published in the scientific literature, most of which are substrate analogs (Appendix A), the only widely used inhibitor is the herbicide glyphosate, which was discovered nearly 50 years ago and acts via the specific inhibition of plant and malarian EPSP synthase (EPSPS) [16,17,18,19].

The shikimate metabolic pathway (Appendix A) is found in plants, algae, bacteria, fungi, and apicomplexan parasites. It involves seven enzymes that catalyze a series of reactions to form chorismate, a precursor of aromatic amino acids, iron-scavenging siderophores, folic acid, vitamin K, and ubiquinone [10,11,12,13,14]. The extreme importance of the synthesis of aromatic amino acids in the bacterial life cycle and their structural as well as kinetic properties have been reviewed in detail by many authors to whom we refer the reader [20,21,22,23,24,25,26,27,28,29].

In the absence of clinically relevant antibacterial drugs against any of these enzymes, there is a legitimate concern that these targets are not sufficiently ligandable and/or druggable. Druggability is defined as the ability of a target to be modulated by potent, small “drug-like” molecules. Drugs that bind to a target that is more druggable are likely to have higher ligand efficiencies, require lower doses and exposures, and have a lower risk of failure during the developmental stages of the drug. To bind drug-like molecules, a binding site must also be ligandable, i.e., it must be able to bind inhibitors with sufficient affinity and be complementary or compatible with their physicochemical properties [30]. 

There are many strategies we can use to assess the ligandability and druggability of targets [31]. One of the most rigorous is the evaluation of existing inhibitors or hit rates of HTS campaigns. It was first introduced in 2006 by Macarron, who published a retrospective analysis of HTS results at GlaxoSmithKline (GSK) [32]. Although many other metrics have subsequently been reported, their precision depends heavily on the number of compounds present [33,34,35]. For example, Vukovic and Huggins [36] described a reward-based metric for quantifying target ligandability for which BindingDB [37,38] reports > 100 Ki values. Since the number of submicromolar inhibitors is insufficient to make a reliable prediction, such an analysis could not be performed with sufficient reliability for most enzymes of the shikimate pathway (Appendix A). 

For proteins with 3D structures that are known or predicted using AI systems such as AlphaFold, several computational methods can be used that provide medicinal chemists with important guidelines for prioritizing targets before they perform costly high-throughput screening [30,36]. Over the years, many crystal structures of all seven enzymes of the shikimate pathway have become available in both apo and liganded forms, making such in silico studies feasible. Although numerous software and online platforms have been developed for this purpose, including FTMap [39], DOGSiteScorer [40], SiteMap [41], Fpocket [42], Sitehound [43], etc., few of these have been used for the analysis of shikimate pathway enzymes. To the best of our knowledge, there are only two studies in which the FTMap server was used to identify hotspots of DHQS from *S. aureus* [44] and SDH from *H. pylori* [45].

However, even if a particular binding site proves to be druggable, it may not be a suitable target for antibacterial drug development because its evolutionary conservation is not high enough. The extent to which the amino acids that constitute a particular target are evolutionarily conserved correlates strongly with its structural and functional importance; it is also a predictor of the likelihood that a potential target will be susceptible to mutation and the development of resistance [46]. Evaluations of evolutionary conservation have already been performed, to some extent, for the enzymes of the shikimate pathway, mostly using multiple sequence alignments (MSA) of sequences from different kingdoms for each enzyme of the pathway [20,47,48,49,50,51,52,53,54,55,56,57,58,59,60,61,62,63,64]. 

Evolutionary conservation and druggability analyses are therefore crucial for the validation of antibacterial targets, and it is surprising that the combination of these two methods has not yet become a standard target validation procedure in the literature. Such procedures are also rare in other antimicrobial fields, for example, in the antiviral field. To our knowledge, there is only one study by Kukol and Hughes in which both features were considered simultaneously in a large-scale analysis of the influenza A virus nucleoprotein sequence [65]. There are also a few reports in which the identification of hotspots or binding sites was performed in parallel with the calculation of their evolutionary conservation [66,67,68]. 

The aim of this study is to perform a combined comprehensive in silico analysis of evolutionary conservation and druggability for the enzymes of the shikimate pathway in pathogenic bacteria. In this context, we intend to identify hotspots with FTMap and binding pockets with SiteMap and FTSite in a representative set of crystal structures of each enzyme of the pathway. Since binding regions are not strictly limited to catalytic sites, the presence of other regions, such as allosteric sites, that may be of equal interest, was also investigated. The available crystal structures were analyzed for this purpose only because the identification of hidden allosteric sites not apparent from the available crystal structures alone would require a large amount of conformational sampling. Such an analysis is, therefore, beyond the scope of this manuscript. The evolutionary conservation of each enzyme and binding site was calculated using ConSurf [69], and the results of each binding pocket were combined with the druggability results calculated using SiteMap. Subsequently, the physicochemical properties calculated by SiteMap were analyzed and compared with the physicochemical properties of available small-molecule inhibitors. To our knowledge, this is the first study of the shikimate pathway that considers evolutionary conservation, the identification of binding pockets, and an evaluation of druggability. We believe that this systematic study provides structural and molecular insights into the druggability and evolutionary conservation of the enzymes of this pathway and predicts whether they are suitable targets for the development of new broad-spectrum antibacterial drugs. This would open new opportunities for the development of novel inhibitors against these enzymes. 

## 2. Results and Discussion

### 2.1. Study Design

In the present study, evolutionary conservation and druggability were investigated for seven enzymes of the shikimate metabolic pathway. For each protein of the pathway, we also analyzed its relevant functional role and the existence of different isozymes in the literature. The results of our analysis can be summarized in several steps that are illustrated in the workflow shown in Figure 1.

To assess evolutionary conservation, the first step was to extract all sequences from the UniProt database and perform a multiple sequence alignment (MSA). This information was then used in the next steps to interpret the identified binding sites and druggability results. In parallel, the FTMap and FTSite servers were applied to selected enzyme structures to identify heatmaps and binding sites (including allosteric sites). This analysis was complemented by the use of SiteMap, which identified binding pockets, calculated key physicochemical properties, and estimated druggability parameters for each binding pocket, such as DScore and SiteScore [41]. The analysis was completed by comparing these properties between representative enzymes of each class and by comparing the physicochemical properties of currently available nanomolar inhibitors of each enzyme. Complete data for all enzymes analyzed, including their physicochemical properties and druggability data for each identified site, can be found in Appendix A. A detailed description of the workflow procedure can be found in Materials and Methods, Section 3.1.

Four enzymes in this pathway, namely DHQS, SDH, EPSPS, and CS, each exist in only one enzyme form. In contrast, the other three enzymes, namely DAHPS, DHQase, and SK, each exist in at least two isozymes. Given that they differ in their physicochemical properties, amino acid sequences, quaternary structures, molecular mass, and occurrence in different bacteria, they were each analyzed differently. Both sequences of DAHPS isozymes were used in the MSA because they share a common 3D structure and were present in 14% of genomes analyzed [70,71,72,73,74,75]. In contrast, the two DHQase families were analyzed separately (DHQase I and II) because there is neither a unique amino acid sequence nor 3D similarity between the two [50,76]. Only the dominant form of SK, namely SK II, was used in the MSA [77]. A more detailed explanation of the properties of each isozyme, with additional details on their exact lengths, can be found in the Appendix A. 

### 2.2. The Identification of Binding Sites 

The binding potential of all seven enzymes in the pathway was predicted using FTMap, FTSite, and SiteMap for a set of 42 proteins from the RCS PDB. In general, for each enzyme, the number of heatmaps identified with FTMap was higher than the number of ligand-binding sites identified with SiteMap, as shown in Figure 2. This is a direct result of differences in the methodology used by both programs to calculate these sites, as well as differences in the basic definitions of heatmaps and ligand-binding sites. Heatmaps, also known as hot spots, are relatively small regions of the binding surface that contribute to a large fraction of the binding energy [78,79]. Fesik and colleagues [35] found that small organic compounds bind almost exclusively to well-defined, localized regions of proteins, regardless of their affinity. The identification of hotspots is therefore important for the study of macromolecule–ligand interactions, the identification of binding sites, the determination of druggability, and the determination of the input for fragment-based ligand discovery (FBLD) [39]. On the other hand, ligand-binding sites are regions that are larger and may contain multiple hotspots [80]. A tight-binding inhibitor may therefore bind to multiple hotspots simultaneously. It is not guaranteed that a high hit rate means that it is easy to identify a small-molecule inhibitor with sufficient binding affinity. Many proteins have binding hotspots that bind fragments with high affinity but are considered difficult drug targets [35,79].

On average, FTMap and SiteMap identified 6.1 ± 1.9 and 10.5 ± 1.5 binding sites and heatmaps, respectively, that were located at the substrate- (orthosteric sites), the cofactor-binding site, or at other parts of the surface (potential allosteric sites) (Figure 2). In contrast to SiteMap and FTMap, where multiple spots were detected, FTSite identified only three binding sites per enzyme. The number of potential heatmaps and binding sites is highly dependent on the shape and physicochemical properties of the cavities on the protein surface. These factors are also incorporated into the equation used by SiteMap to calculate the two druggability scores, SiteScore and DScore (see the experimental section) [39,41]. For most enzymes, the number of binding sites identified by SiteMap depends on the molecular weight of the enzymes and also correlates positively with the number of heatmaps identified by FTMap. An exception is CS, for which SiteMap identified additional binding sites in the C-terminal part of the enzyme that were not identified by FTSite.

### 2.3. Evolutionary Conservation

Figure 3 shows the projections of conservation scores obtained after MSA of sequences from different pathogenic bacteria. The scores were projected onto the protein structure of each of the seven enzymes and classified by the ConSurf server into classes ranging from 9 (highest conservation) to 1 (highest variability). The most variable positions (grade 1) are colored turquoise, the moderately conserved positions (grade 5) are colored white, and the most conserved positions (grade 9) are colored maroon. We must clarify that a conservation score calculated by ConSurf is a relative measure of evolutionary conservation at each sequence position and does not necessarily mean that the highest score indicates 100% conservation. These figures provide an overview of the structure and surface area of these enzymes and also illustrate the regions with highly conserved residues. In general, most of the conserved residues are located inside the enzyme at the substrate-binding site. This is no surprise, as the degree of evolutionary conservation is often indicative of the importance of the position in maintaining the structure and/or function of the protein. Moreover, it has already been reported that selective pressure acting on the substrate-binding site generally spreads to the rest of the protein via residue–residue contacts [81]. Thus, a gradient of conservation can be observed, such that the closer a residue is to the binding site, the more conserved it is, which is particularly observed in surface representations of enzymes (B and C). Small areas of conserved residues can be observed in the other parts of each protein, but they are small and therefore not important from a drug design perspective. An exception is CS, where a larger conserved region is found at the C-terminal end of the enzyme.

Next, we analyzed the percentage of conserved and highly conserved residues (scores 8 and 9) in each enzyme, structural motif, and binding pocket. The results are shown in Figure 4. The first histogram (A) shows that most of the conserved residues are located on the α-helices, which are inside the enzyme. Exceptions are DAHPS, where most conserved residues are found on the loops, and CS, where these residues are evenly distributed between helices and loops. 

A second, stacked histogram (B) shows that all enzymes in the shikimate pathway are evolutionarily well conserved, with conservation scores of 8 and 9 ranging from 20% to 45%, consistent with their functional importance to the bacteria. The evolutionarily least conserved are DAHPS and DHQase I, which have 23% and 21% conserved residues, respectively. The lower conservation of DAHPS is not surprising because the sequences taken for MSA were distributed between both enzyme classes and have only 10% sequence identity [70,71,72]. On the other hand, CS is the most evolutionarily conserved of all enzymes in the shikimate pathway and contains 44% of residues with high conservation scores.

This is also partially evident from the third histogram (C), which shows the evolutionary conservation of binding sites. These results indicate that substrate- and cofactor-binding sites have significantly higher conservation scores compared to other binding sites. It is also clear that substrate-binding sites in the pathway have a similar proportion of residues, with the highest conservation scores of 8 and 9 per binding site, ranging from 60% to 90%. In contrast, for the binding sites that do not bind substrates, this proportion is only 20 to 40%. The observation that substrate-binding sites are more conserved compared with the rest of the protein is consistent with previous literature reports [82,83]. These results also suggest a possible explanation for the high evolutionary conservation of CS. Based on the last histogram, the substrate-binding site of CS is not significantly different from other enzymes in this pathway in terms of the percentage of highly conserved residues. Its overall higher evolutionary conservation is a consequence of the fact that other parts of the enzyme that do not bind the native ligand, particularly at the C-terminal end of the enzyme, are somewhat more conserved in CS than in other enzymes of the pathway, resulting in the higher evolutionary conservation of the entire CS enzyme. 

Because three enzymes in this metabolic pathway, namely DHQS, SDH, and SK, require cofactors for their normal function, we further analyzed whether there is a difference in evolutionary conservation between cofactor- and substrate-binding sites. Unfortunately, we could not use the pre-existing residues from the SiteMap results because, in most cases, the cofactor- and substrate-binding sites collided and formed one large binding site. Therefore, residues within 4 Å of the cofactor or substrate were selected for the analysis. A list of the residues of the selected representatives for each enzyme can be found in the Appendix A, along with the ConSurf score and the maximum identity of each amino acid. It was found that although both the cofactor- and substrate-binding sites have high conservation scores and are therefore viable targets for antibacterial drugs, there is a significant difference in evolutionary conservation between the two sites. The cofactor-binding sites are, on average, 20–30% less conserved than the substrate-binding sites, which all contain 100% residues with scores of 8 and 9. These results, along with the potential selectivity issues arising from the fact that these cofactors are used by several other enzymes in the human body, suggest that the substrate-binding sites of DHQS, SDH, and SK are the preferred primary targets from a drug design perspective. 

The only enzyme in the entire pathway reported to be controlled by allosteric feedback inhibition is DAHPS. However, in most organisms, this enzyme exists in three isozymes: aroF, aroG, and aroH, each regulated by different aromatic amino acids, namely Tyr, Phe, and Trp [84,85,86,87,88]. Because all three isozymes may be present simultaneously in the bacterial cell, a potential antibacterial drug would need to act simultaneously on all three isozymes with similar binding affinities. Our results show that while the orthosteric site of this enzyme is highly conserved, the allosteric site is highly variable. For example, in *E. coli*, the only conserved amino acid was S211, with a conservation value of 8, and all other residues were significantly less conserved (Appendix A). Although it is known that allosteric sites are evolutionarily less conserved because they have not been subject to direct evolutionary pressure to conserve important functional residues [89], this can be an advantage in some cases because it leads to improved selectivity and is desirable when only one species is targeted [90,91]. For example, allosteric sites of functional proteins of *M. tuberculosis* (Mtb) have been used in the study of the Mtb enzymes ornithine acetyltransferase [92] and tryptophan synthase [93]. Thus, this lower evolutionary conservation of all sites is an advantage when only one species is targeted. On the other hand, targeting allosteric sites can also lead to species differences that may hinder the broad-spectrum efficacy of a drug in different pathogenic bacterial species [90,91].

### 2.4. Druggability Assessment

Both ligandability and druggability mappings were examined in detail for each of the 42 proteins from the RCS PDB database using all three programs. Both FTSite and FTMap ranked the consensus clusters by the number of unbound contacts between the protein and all probes in the cluster. Representative structures of each enzyme were then superimposed, and the three highest-scoring clusters from the FTMap and FTSite analyses were selected and are shown in Appendix A. Although both FTMap and FTSite are of great help in prioritizing binding sites and are also widely used for druggability assessment [33,36,94,95,96], we decided to use SiteMap for this purpose because it has demonstrated its effectiveness in several publications and has also been extensively validated on a set of 538 crystal structures [33,66,97,98,99,100,101,102,103,104,105,106,107]. The results of FTMap and FTSite were therefore complemented by the SiteMap analysis of the same PDB structures. 

In general, the highest-ranked ligandable regions selected by all three programs were the orthosteric and the cofactor regions. The exception was DAHPS, where the heatmaps were distributed around the enzyme, but most of the highest-ranking clusters were in the orthosteric region regardless of the X-ray structure used. These results were also confirmed by DScore and SiteScore calculations. 

Figure 5 shows that the substrate-binding sites are generally much more ligandable than the allosteric binding sites. Two black dotted lines divide the scatter plot into four regions, with DScore and SiteScore values above and below 0.8, respectively; this is a threshold proposed by SiteMap developers to identify sites that are ligandable (SiteScore) and/or druggable (DScore) [80]. Based on these results, the majority of allosteric sites have both scores below 0.8 and are located in the lower-left quadrant, indicating the least promising sites. The majority of orthosteric sites and the minority of allosteric sites have scores above 0.8, indicating their druggability. There is also a smaller number of sites in the lower right quadrant that are undruggable but have the potential to bind small molecules with high affinity.

The observed ability of all three programs to predict substrate- and cofactor-binding sites as equally favorable for noncovalent binding also suggests a possible targeting strategy in which an inhibitor targeting the substrate-binding region could readily extend to the cofactor-binding region and interact more strongly with surrounding amino acids.

Because one of the goals of our study was to identify binding sites other than substrate- or cofactor-binding sites, the results from FTSite and SiteMap were compared. To reduce the number of sites that could not be targeted, a threshold of 0.8 was set for SiteScore and DScore, as suggested by the SiteMap developers. This proved to be very useful because most of the undruggable binding sites were filtered out, leaving only 33% of the original sites [80]. We also found that most of the highly conserved cavities on the surface of the C-end of CS, which we mentioned earlier, were not druggable enough. Although the evolutionary conservation of protein cavities is often related to their druggability [108], most of these sites were either too small or not sufficiently enclosed. These sites were therefore filtered out and not analyzed further.

As expected, both methods largely agreed on the highest scoring sites (Appendix A). The binding sites that scored higher with the FTSite method largely overlapped with the binding sites that scored higher with SiteMap. However, each method also found binding sites in regions that did not bind a substrate or cofactor and that were not predicted by the other method. For example, for SDH, we identified a potential binding site in 3 of 6 crystal structures that were predicted to be druggable by SiteMap but were not identified by FTSite (Appendix A). In many cases, binding sites observed in one crystal structure and ranked highly by one of the methods were not observed in another structure of the same enzyme belonging to a different bacterial species. This observation could be a consequence of different protein conformations or differences in protein sequences leading to the lower evolutionary conservation of a single protein. Therefore, it was very difficult to decide which of these mismatched binding sites was a prospective drug binding site when only druggability results were considered.

In the context of this work, it was therefore of great importance that the druggability results were examined in conjunction with the evolutionary conservation calculations, which were very helpful in assessing whether a region of a protein was a prospective binding site or not. After using 50% of the conserved residues in a pocket as a threshold, we found that 20% of the original sites were both druggable and evolutionarily conserved, of which only two sites (SDH and CS) did not bind a substrate or cofactor. Further visual inspection revealed that the reason for the higher evolutionary conservation of these two binding regions was partial overlap with the evolutionarily well-conserved substrate-binding region and that both regions were identified as part of a larger pocket in other crystal structures of the same enzyme. The drawback of this approach, however, is that cavities that could serve as potential allosteric sites were likely filtered out because of the well-documented lower conservation of such sites [109,110].

From this, we can conclude that for all seven enzymes studied, only the substrate- or cofactor-binding regions are suitable broad-spectrum antibacterial targets and that no allosteric site can be used for this purpose.

### 2.5. Evaluation of Physicochemical Properties of Binding Sites

The ligandability and druggability of binding sites depend on many physicochemical parameters, such as volume, enclosure, hydrophobicity, and percentage of charged residues [80]. Drug binding pockets tend to be deeper, larger, and more complex in shape; they also tend to have high hydrophobicity, lower hydrophilicity, and high enclosure; the latter maximizes the surface area to volume ratio. The opposite is true for “hard to drug” and undruggable pockets [31,34,40,111]. Since DScore is calculated directly from the number of site points, the enclosure, and the polarity, we can further explain the differences between the druggability of orthosteric and cofactor-binding sites and the nonbinding sites by comparing their basic physicochemical properties.

These properties, which were used for evaluation, are listed in Appendix A. The substrate-binding sites of these enzymes are significantly larger and more enclosed compared to the allosteric sites (*p* < 0.05), which increases their DScores and SiteScores. The size of the pocket is an important descriptor because small cavities, usually consisting of <14 residues, do not promote adequate interaction between the target and the ligand [111,112,113]. On the other hand, substrate-binding sites were found to be more polar compared to allosteric sites (*p* < 0.05), which negatively affects their druggability. This result was not surprising since the substrates and products of each step of the pathway are also highly polar. Our results on the differences in druggability between the orthosteric and allosteric binding sites are consistent with data found in the literature [114,115].

Druggability can also be estimated from other parameters calculated indirectly from SiteMap, such as the proportion of charged residues in a binding site. It is clear from the literature that highly charged pockets are much less druggable than uncharged pockets [31,34,40,111]. Perola, therefore, proposed a threshold of 26.3% to distinguish between druggable and non-druggable pockets [111]. However, although substrate-binding sites were found to be more polar than allosteric sites, we did not detect a significant difference in the proportion of charged residues between the average orthosteric and allosteric sites (*p* < 0.5).

Most of the metrics used to evaluate the potential allosteric sites, such as DScore, the number of site points, the enclosure, and the volume, were below the thresholds that distinguish druggable from the undruggable sites defined in the literature [33,66,97,98,99,100,101,102,103,104,105,106,107]. We can, therefore, additionally confirm that we were unable to identify any of the allosteric pockets as candidates for drug targeting in any of the seven enzymes analyzed.

### 2.6. Druggability of the Substrate-Binding Sites

Next, we turned to a detailed evaluation of the orthosteric and cofactor-binding sites and an analysis of their physicochemical properties. We found that the binding sites of all seven enzymes are ligandable, as indicated by their high SiteScore values around 1; therefore, all are considered particularly promising targets for small molecules (Figure 5). The highest-scoring enzyme in terms of its mean SiteScore value was EPSPS (SiteScore = 1.13), mainly due to its high enclosure, and the lowest-scoring enzyme was DHQase I (*p* < 0.05). Although DAHPS is particularly promising for small molecule development, SiteMap classified it as difficult to drug because its DScore was not significantly different from 0.8 (*p* < 0.05). This does not mean that the site cannot bind ligands, but that it would be difficult to find high-affinity drug-like ligands for it. Other enzymes in the pathway were found to be druggable, with a DScore around the value of 1, and they are, therefore, particularly promising targets for the development of drug-like molecules.

### 2.7. Physicochemical Properties of Substrate-Binding Sites

To explain the lower druggability of DAHPS and to further investigate the binding sites of all other enzymes in the pathway, the physicochemical properties of each enzyme were next evaluated. Such an analysis provides medicinal chemists with important guidelines for predicting the physicochemical properties of potential drugs for the specific binding pocket. The main properties of the individual enzymes are shown in Figure 6.

In general, the greatest differences between the enzymes were found when the volumes of the binding sites were compared. For DHQS, SDH, and CS, the volume is significantly higher than the lower limit of the average submicromolar binding sites, which is a consequence of the additional space occupied by a cofactor rather than by a substrate alone. In contrast, the binding site of SK, which uses ATP as a cofactor, was found to be similar in size to other enzymes that do not require a cofactor.

We also found that most binding sites have enclosures and strengths of van der Waals interactions with the site points (contact) comparable to the characteristics of the average tight-binding site. One enzyme that stands out when considering both properties is EPSPS, which has a very high enclosure and forms strong contacts with its binding site. These two properties increase the likelihood that an inhibitor interacts strongly with the binding site, increasing its ligandability and druggability.

Substrate- and cofactor-binding sites were found to have a higher hydrophilic character than the average druggable tight-binding site. This property is of great importance because it negatively affects the druggability of a binding pocket and also indirectly predicts the polarity of a potential ligand [30]. The higher polarity of the binding site is related to the relatively higher polarity of the substrates of these enzymes. In addition, two enzymes, DAHPS and EPSPS, tend to have an extremely hydrophilic character as a result of their highly charged substrates with two acidic groups. However, unlike EPSPS, which has a very high enclosure that increases its DScore, the enclosure of DAHPS is slightly below the average of tight-binding sites. Consequently, its DScore is at the threshold of 0.8 and it has a very high percentage of charged residues, well above the proposed threshold of 26.3%. This site should therefore be considered an undruggable drug target. Interestingly, although the binding site of EPSPS is also very polar, the percentage of charged residues is below the proposed threshold. The high SiteScore suggests that this binding site is more ligandable compared to other sites in the pathway. On the other hand, a hypothetical submicromolar inhibitor targeting such a highly polar binding pocket should also have higher polarity, which could lead to problems in pharmacokinetic properties, such as a shorter elimination half-life [116].

From a drug-design perspective, a binding site should also have a significant number of hydrophobic regions in a pocket that can bury the drug in the pocket. Since electrostatic interactions and desolvation energies act in opposite directions, their contribution to binding potency is negligible. Therefore, hydrophobic interactions are a driving force in the interactions between a ligand and the binding site and, consequently, have the greatest influence on the strength of drug binding [80,117]. Despite their higher proportion of hydrophilic surface area compared with the average tight-binding site, we found that DHQase I, II, and CS have larger regions of hydrophobic amino acids than other enzymes, and their calculated hydrophobic character is comparable to that of the average drug binding site. For these enzymes, it should be easier to design a drug-like inhibitor that interacts with both hydrophobic and hydrophilic regions of the enzyme.

Because the physicochemical properties of each binding site predict the optimal properties of a potential tight-binding inhibitor, a comparison was made between SiteMap results and the values calculated for published inhibitors. Although the number of drug-like nanomolar inhibitors is too small to predict the druggability of each enzyme in the pathway, a comparison between the predicted properties and the existing inhibitors may provide some clues on how to modify the properties of the inhibitors to further improve their binding efficacy. Therefore, two physicochemical properties, topological polar surface area (TPSA) and hydrogen bond donor/acceptor ratio (don/acc), were calculated for the published inhibitors. These results are shown in Figure 7. Since most of the published inhibitors are derivatives or analogs of native ligands, a slight correlation between their TPSA and the hydrophilic character of the binding sites was expected and, for the most part, found. The inhibitors of highly polar binding sites of EPSPS and DAHPS, therefore, have high TPSA values, whereas the inhibitors of other enzymes have similar TPSA values. As expected, EPSPS and DAHPS inhibitors are not drug-like due to their high polarity, which is a consequence of the presence of phosphonic and carboxylic groups in these compounds. Therefore, the binding strength and the drug-likeness of the inhibitors could be further improved by changing the ratio of don/acc groups and eliminating at least one of the highly acidic groups or by preparing prodrugs. In the boxplot in Figure 7, we see that the predicted don/acc ratio of an ideal inhibitor is higher than the actual ratio for the DAHPS, DHQS, and CS enzymes. Thus, an inhibitor with a higher don/acc ratio would be a better match for these binding sites.

## 3. Materials and Methods

### 3.1. Study Design

To assess evolutionary conservation, the first step was to extract all sequences and other data for each sequence from the UniProt database. All filtering and analyses of the data were performed using KNIME [118], and only representative sequences from pathogenic bacteria were selected. Subsequently, MSA of the selected sequences was performed, and a phylogenetic tree (PT) was constructed. MSA and PT are prerequisites for the calculation of evolutionary conservation, which was performed using the free online bioinformatics tool ConSurf server [69,119,120,121]. This information was then used in the next steps to interpret the identified binding sites and druggability results. Because many X-ray structures are publicly available, the 3D structures of each bacterial protein were next extracted from RCS PDB. The structures of each enzyme were then aligned, and root-mean-square deviation (RMSD) was calculated, followed by k-means clustering. Representative structures from each of the six clusters were selected based on resolution, average B-factor, and the number of missing residues in each PDB structure. Structures were then prepared using the Protein Preparation Wizard implemented in the Schrödinger Suite, which is described in more detail in Section 3.4. To identify all heatmaps and binding sites (including allosteric sites) in each protein, the FTMap and FTSite servers were applied to selected representatives. This analysis was complemented by the use of SiteMap, which uses the interaction energies between the protein and grid probes to locate energetically favorable sites, identify binding pockets, calculate key physicochemical properties, and estimate druggability parameters, such as DScore and SiteScore, of each binding pocket [41]. The analysis was completed by comparing these properties between representative enzymes of each class and by comparing the physicochemical properties of the currently available nanomolar inhibitors of each enzyme. Complete data for all enzymes analyzed, including their physicochemical properties and druggability data for each identified site, are provided in Appendix A.

### 3.2. Identification of Hotspots and Binding Sites with FTMap and FTSite

FTMap [39] is a computational mapping server that identifies binding hotspots of macromolecules using 16 small organic probe molecules of different sizes, shapes, and polarities distributed on the molecular surface. Hotspots are smaller regions of proteins that contribute significantly to the binding of a drug to the binding site, and their strength determines the druggability of a site. They are very important for fragment-based ligand discovery (FBLD), as the ligand moieties that interact with them are essential for binding. FTSite is a second server that identifies ligand binding sites by using consensus sites determined via the FTMap server to identify and rank binding sites. Similar to FTMap, it uses molecular probes to map the macromolecular surface and finds the most favorable positions for each probe type. Probes are then clustered and ranked by the number of nonbonded contacts between the protein and all probes in the consensus cluster rather than by the number of probe clusters. The following probes are used by each server: Acetamide, acetonitrile, acetone, acetaldehyde, methylamine, benzaldehyde, benzene, isobutanol, cyclohexane, *N*,*N*-dimethylformamide, dimethyl ether, ethanol, ethane, phenol, isopropanol, and urea [39].

All prepared PDB files were uploaded to the FTMap and FTSite servers for mapping. Upon completion, the results were downloaded as a Pymol [122] session file (.pse) for further analysis. Amino acids within 4 Å of each cluster were selected and exported in .pdb format. PDBest [123], a freely available platform for the manipulation, filtering, and normalization of biomolecules, was used to extract residue numbers that were further processed using KNIME [124].

### 3.3. Druggability Assessment with SiteMap

SiteMap [41] is software used for binding site identification and evaluation; it is implemented in Schrödinger’s Maestro [85]. We chose SiteMap because it has been shown to be effective in a number of publications on various proteins [33,66,97,98,99,100,101,102,103,104,105,106,107]. A comprehensive validation of SiteMap was published using 538 crystal structures from the PDBbind database [80]. In this study, SiteMap accurately identified 86% of the highest-scoring binding sites. In addition, it was found that the size of the binding site, as measured by the number of site points found, and the relative openness of the site, as measured by the exposure and enclosure properties, were the most important terms for distinguishing binding sites from nonbinding sites.

SiteMap uses the interaction energies between the protein and the grid probes to find energetically favorable sites. In a three-step process, a program generates a grid of points (site points) and then uses their energetic properties to score the sites. During this process, various physicochemical properties are calculated: size, volume, degree of enclosure or exposure, degree of contact, hydrophobic or hydrophilic character, and their balance and hydrogen bonding properties (acceptors/donors), all of which are listed in Table 1 along with their reference values for average submicromolar inhibitors. Exposure and enclosure provide different estimates of how open a binding site is to the solvent. The lower the exposure value and the higher the enclosure, the better. The hydrophobic/hydrophilic balance measures the relative hydrophobic and hydrophilic character of the site. The donor/acceptor character indicates the extent to which a potential ligand can donate rather than accept hydrogen bonds [80].

In addition, SiteScore and DScore were also calculated to evaluate overall ligandability and druggability properties.

SiteScore predicts whether a binding site could be a drug binding site. The developers suggest a value of 0.80 to distinguish sites that bind ligands from those that are not known to do so. Thus, a value greater than 1 indicates a particularly promising site, as seen in Equation (1) [80].
(1)$$\mathrm{SiteScore}=0.0733\sqrt{n}+0.6688e-0.20p$$
where 

n = the number of site points found for the site, capped at 100;

e = the degree of enclosure score;

p = the hydrophilic score computed for the site, capped at 1.0 to limit the impact of hydrophilicity in charged and highly polar sites.

DScore is a druggability score that includes terms that promote ligand binding but offsets them with a term that penalizes increasing hydrophilicity. Again, the number of site points is limited to 100, but the score for hydrophilicity is not capped. See Equation (2).
(2)$$\mathrm{DScore}=0.094\sqrt{n}+0.60e-0.324p$$
where

n = the number of site points found for the site, capped at 100;

e = the degree of enclosure of the site;

p = the hydrophilic score computed for the site.

To facilitate the comparison of the three-dimensional structures of all seven enzymes in the shikimate pathway, the prepared proteins were first aligned using the Schrödinger’s Protein Structure Alignment utility in Maestro [125]. Proteins were then prepared according to the procedure described below. SiteMap was applied to all selected enzymes to identify up to 10 potential binding sites with the highest score. Sites were retained if they comprised at least 15 site points per reported site. The narrower definition of hydrophobicity was used, along with a fine grid (0.35 Å). Site maps that were 4 Å or more from the nearest site points were truncated. The analysis of SiteMap calculations was performed using KNIME [124], extracting physicochemical property data and residue numbers for further processing.

### 3.4. Protein Acquisition and Preparation

Selected 3D structures were imported from the Protein Data Bank into Maestro [125]. To prevent regions between protein monomers from forming unphysical sites that give good results but exist only in the crystal lattice and not in solution, only one monomer chain was retained, usually chain A. Structures were then prepared using the Protein Preparation Wizard implemented in the Schrödinger Suite [125]. In each PDB structure, ligands, waters, and other co-crystallized molecules were removed except for the Mg cofactor, if present. Bond orders were automatically assigned, hydrogens were added, selenomethionines were converted to methionines, missing side chains were added, ligands were removed, disulfide bridges were created when possible, waters beyond 5 Å radius of heteroatoms were added, and heteroatoms were protonated at pH 7.0. The impref utility was used to perform constrained minimization of the protein with a maximum root-mean-square deviation (RMDS) of 0.30 Å.

### 3.5. Superposition of 3D Structures and Selection of Representative Crystal Structures

All 3D structures available for each enzyme were taken from the Protein Data Bank (PDB) and prepared using the procedures described in Section 2.3. Water was removed, and only one chain, mostly chain A, was retained for analysis. All structures were then aligned, and the RMSD was calculated for each pair. The values were exported to KNIME in the form of a 2D matrix for further processing. Only bacterial enzymes without induced residue mutations were selected, and then the k-means algorithm was used to generate six representative clusters. The selection of the representative structure from each cluster was then based on resolution and the extent to which some protein chains were incomplete.

### 3.6. Multiple Sequence Alignment

The protein sequences of the individual enzymes of all species were obtained from UniProt [126]. Data on the total number of sequences extracted per enzyme and the distribution among kingdoms are given in Appendix A. Outliers that had only 60% residues compared with the average length of the protein family were filtered out, and then a unique representative was selected for each human pathogenic bacterium. A list of the number of human pathogenic microorganisms was obtained from the Kyoto Encyclopedia of Genes and Genomes [127].

The protein sequences of the human pathogens were then subjected to multiple sequence alignment with MuscleWS in Jalview software [128,129] using the default settings. After converting the alignment to Fasta file format, a phylogenetic tree was calculated based on the neighbor-joining method using similarity scores calculated by the blocks substitution matrix 62 (BLOSUM 62), which measures the evolutionary relationship between each pair of sequences in the alignment [130]. After converting the MSA and phylogenetic tree to the Clustal Alignment (.aln) and Newick (.txt) format files, respectively, the multiple alignment file and phylogenetic tree were subsequently used by the ConSurf server [69,119,120,121].

### 3.7. Calculation of Evolutionary Conservation

The ConSurf server [69,119,120,121] is a bioinformatics tool for estimating the evolutionary conservation of amino/nucleic acid positions in a protein/DNA/RNA molecule based on the phylogenetic relationships between homologous sequences. The program uses either an empirical Bayesian method or a maximum likelihood method [131,132] to estimate the evolutionary rate of the protein sequence, which is then projected onto the 3D structure of a selected enzyme.

Both of the .aln files from the MSA and the phylogenetic tree were used by the ConSurf server to calculate the conservation scores, which were then projected onto the X-ray structure of the selected enzyme. The evolutionary conservation of the selected enzymes in pathogenic bacteria was then calculated using the empirical Bayesian calculation method [131,132] with a default substitution model. The conservation scores calculated by ConSurf are a relative measure of evolutionary conservation at each sequence site in the target chain. The obtained continuous conservation scores were mapped onto the X-ray structure of a protein and divided into a discrete scale of nine grades for visualization, from the most-variable positions (grade 1), which are colored turquoise, through intermediately conserved positions (grade 5), which are colored white, to the most-conserved positions (grade 9), which are colored maroon. Results were downloaded from the ConSurf website as a .pdb file containing data for 3D coordinates and conservation scores. Figures mapping the conservation scores to the protein structures were subsequently generated using Pymol [122] for one representative of each enzyme and are shown in Figure 6. A list of ConSurf grades was also obtained and imported into KNIME to extract the names of all residues, along with their conservation scores and identity thresholds.

### 3.8. Calculation of TPSA

Information on inhibitors for each enzyme was obtained from the literature and the ChEMBL database [133]. Inhibitors were first prepared using LigPrep, which is implemented in the Schrödinger Suite [125]. Conformations of the molecules were generated using the OPLS4 force field and ionized using Epik [134,135] at a target pH of 7 ± 2. The molecules were desalted, and the tautomers were generated. The TPSA was then calculated using QikProp, which is also implemented in the Schrödinger Suite [125].

### 3.9. Statistical Analysis

To compare the means of DScore and SiteScore between samples, the Kolmogorov–Smirnov test was used to test whether the data were normally distributed, followed by the Mann–Whitney U test. The difference between the two groups was statistically significant when the *p*-value was less than 0.05.

### 3.10. Generation of Figures

The graphical representation of the three-dimensional structures of proteins was performed using Pymol [123]. Boxplots and stackplots were generated using the Seaborn [136] and Matplotlib [137] libraries in Jupyter notebook [138].

## 4. Conclusions

The exploitation of important but underexplored antibacterial targets is one of the ways to address the urgent need for new antibacterial agents. In the absence of clinically relevant antibacterial drugs against any of the enzymes of the shikimate pathway, we hypothesized that these targets are simply not ligandable or druggable enough to produce a drug-like inhibitor against any of them, given the 70 years of research in this field.

In the present study, we report an in silico assessment of the evolutionary conservation, ligandability, and, most importantly, druggability of the enzymes belonging to the shikimate pathway. To our knowledge, this is the first and most comprehensive study using a combination of all three features for all of these enzymes.

Because mutations and resulting drug resistances play an important role in antibacterial drug development, a validation of antibacterial targets should involve a combination of estimations of druggability, evolutionary conservation, and physicochemical properties of individual binding sites. Using this approach, we mapped the conserved residues in the 3D structures of individual enzymes using ConSurf and applied these results to the assessment of discovered binding pockets; this was performed using FTMap, FTSite, and SiteMap. We have shown that only substrate-binding sites in various pathogenic bacteria are both evolutionarily conserved and ligandable. With the exception of DAHPS, these sites can be targeted by drug-like molecules and are all attractive targets for pharmacological modulation. All of these sites have higher polarity than the average tight-binding site, with DAHPS and EPSPS having particularly high polarity. Because the physicochemical properties of a binding site influence the properties of inhibitors targeting the site, we predict that any drug targeting EPSPS would also be highly polar, which was also observed in the analysis of existing inhibitors targeting EPSPS.

Although the focus of our workflow presented here was exclusively on the development of broad-spectrum antibacterial agents, the same approach could easily be applied to the analysis of the same enzymes as narrow-spectrum antibacterial targets. For example, 42% of all identified sites that are not orthosteric were found to be druggable. Because the evolutionary conservation of protein cavities is often related to their druggability, at least some of these cavities may be conserved in one bacterial species and not in the others.

We believe that this systematic study provides a sufficient structure-based rationale to accelerate the exploration of these underexplored targets. In addition, our workflow presented here may open new opportunities for the development of new antibacterial drugs targeting antibacterial agents that are not strictly related to the shikimate pathway.

## Figures and Tables

**Figure 1 antibiotics-11-00675-f001:**
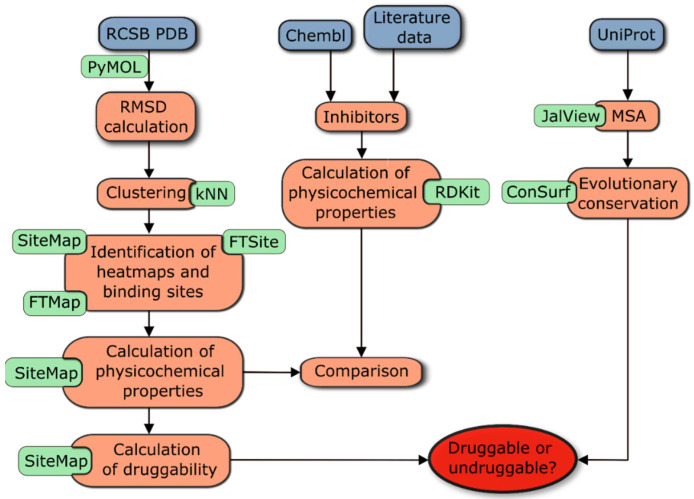
Summary of the workflow to determine the druggability, evolutionary conservation, and physicochemical properties of binding pockets of enzymes in the shikimate pathway.

**Figure 2 antibiotics-11-00675-f002:**
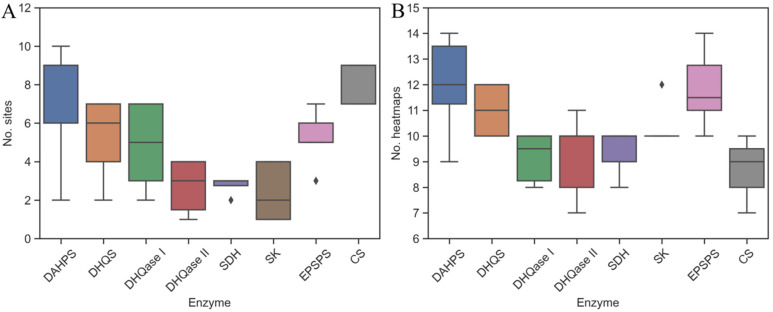
Identification of binding pockets: (**A**) SiteMap and (**B**) FTMap each identified several binding sites and clusters per enzyme.

**Figure 3 antibiotics-11-00675-f003:**
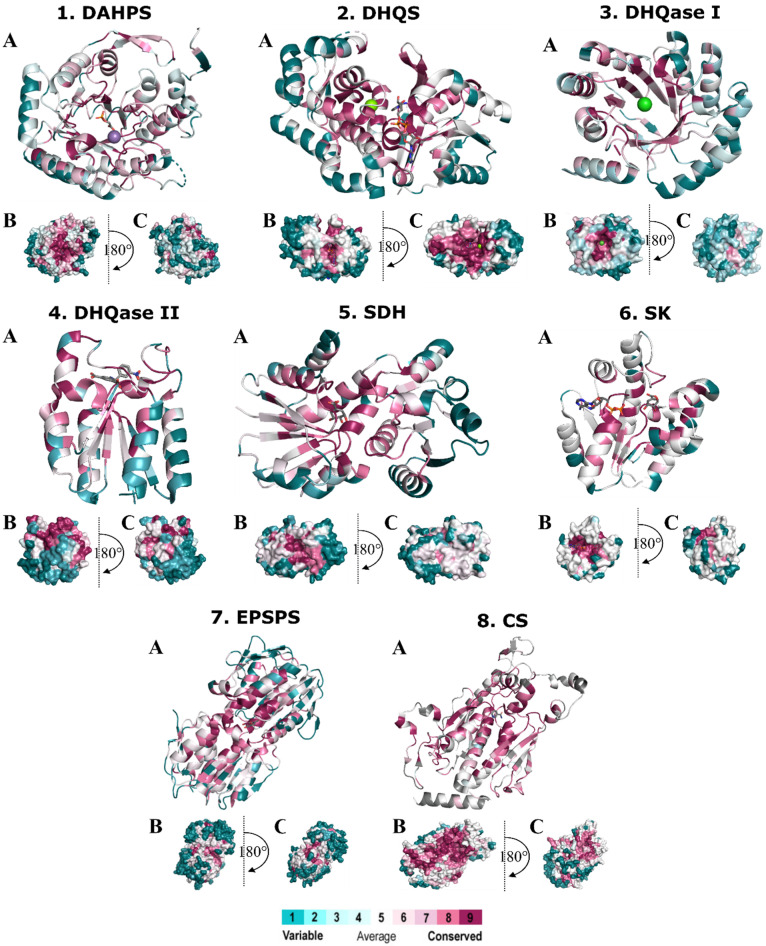
Spacefill (**A**) and surface (**B**,**C**) representation of the representative X-ray monomer structures of the individual enzymes in the shikimate pathway with the degrees of conservation mapped onto the monomer structure. Amino acids are colored according to their degree of conservation using the color-coding bar, with turquoise-through-maroon indicating variable-through-conserved residues. Enzymes: 1. DAHPS from *E. coli* (PDB: 1GG1, ligand: 2-phosphoglycolic acid), 2. DHQS from *A. baumannii* (PDB: 5EKS, ligand: NAD), 3. DHQase I from *S. enterica* (PDB: 4CNN), 4. DHQase II from *M. tuberculosis* (PDB: 4KIW, ligand: 5-[(3-nitrobenzyl)amino]benzene-1,3-dicarboxylic acid), 5. SDH from *M. tuberculosis* (PDB: 4P4G, ligand: shikimic acid), 6. SK II from *M. tuberculosis* (PDB: 2IYQ, ligand: shikimic acid, NAD), 7. EPSP synthase from *E. coli* (PDB: 2AA9, ligand: shikimic acid), 8. CS synthase from *M. tuberculosis* (PDB: 2QHF, ligand: nicotinamide).

**Figure 4 antibiotics-11-00675-f004:**
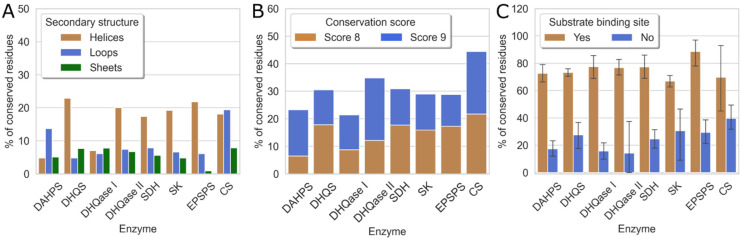
The proportion of evolutionarily conserved residues for individual enzymes. (**A**) Distribution of grade 8 and 9 residues among α-helices, β-sheets, and loops; (**B**) distribution of residues with conservation scores 8 and 9; and (**C**) binding sites identified by SiteMap.

**Figure 5 antibiotics-11-00675-f005:**
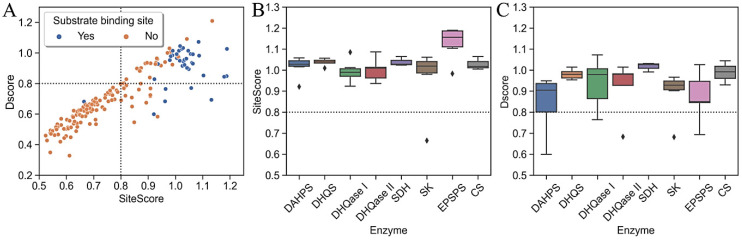
Distribution of ligandability and druggability parameters. (**A**) Scatter plot of DScore versus SiteScore numbers per type of binding site. Two black dotted lines indicate a boundary where both scores are 0.8, dividing the scatterplot into four regions, each with a different potential for druggability and ligation; (**B**) DScores and (**C**) SiteScores of substrate-binding sites per each enzyme in the pathway. Dotted lines in each boxplot indicate a reference value for the average submicromolar site.

**Figure 6 antibiotics-11-00675-f006:**
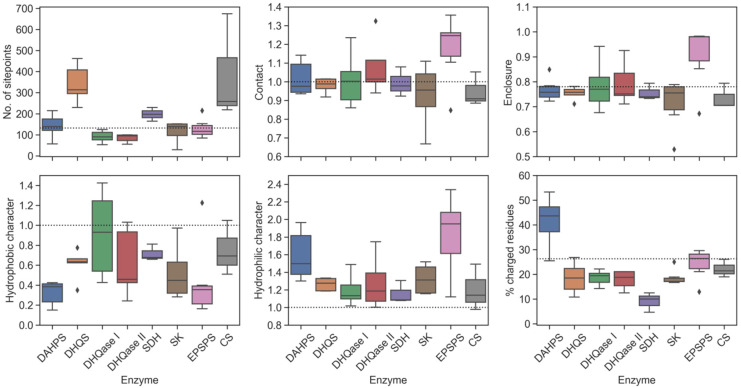
Key physicochemical properties of each enzyme in the shikimate pathway. The dotted line in each boxplot indicates a reference value for the average submicromolar site.

**Figure 7 antibiotics-11-00675-f007:**
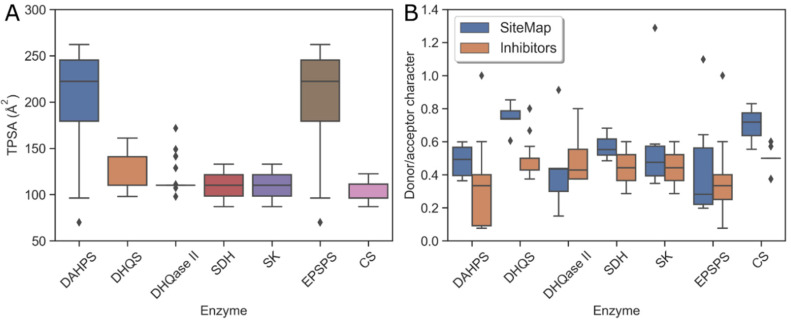
Physicochemical properties of the substrate-binding sites and the available inhibitors. (**A**) Median TPSA of inhibitors per enzyme; (**B**) proton donor/acceptor character of inhibitors compared to binding sites per enzyme.

**Table 1 antibiotics-11-00675-t001:** Properties calculated by SiteMap [41] along with the average property score for a submicromolar site. Data from [80].

Property	Lower Limit ^1^
Number of site points	132
Site score	1 (0.8)
DScore	1 (0.8)
Exposure	0.49
Enclosure	0.78
Contact	1.0
Phobic, Philic ^2^	1.0
HL balance ^3^	1.6
Donor/acceptor	0.76

^1^ Numbers are calculated for an average submicromolar site; ^2^ Phobic, Philic = Hydrophobic property, Hydrophilic property; ^3^ HL balance = Hydrophilic/lipophilic balance.

## Data Availability

Not applicable.

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
