# Peer review of "An Evolutionary Conservation and Druggability Analysis of Enzymes Belonging to the Bacterial Shikimate Pathway"

_antibiotics, 2022, doi:10.3390/antibiotics11050675_

Round 1

Reviewer 1 Report

Summary

The author uses a combination of in silico analyses to evaluate the enzymes involved in bacterial shikimate biosynthesis as potential antimicrobial drug candidates. More specifically, by evaluating both evolutionary conservation and druggability, the author identified the substrate binding sites of each of these enzymes with the exception of DAHPS as viable drug targets.

General

Detailed in silico analyses on alternative antimicrobial targets such as those involved in the shikimate pathway are certainly of interest given the antibiotic resistance crisis. I would’ve liked to have seen some proof of concept biochemical data to support the in silico predictions, but probably beyond the scope of this study.

Specific Comments:

  • Figure 2 – SK (Shikimate Kinase) inhibitor should have a flat line and not a wedge for the carboxylic acid group (no stereochemistry here)
  • Could mention AlphaFold at some point in the intro – this will no doubt serve as an important tool for drug discovery
  • Figure 4 – Comparisson spelt incorrectly?
  • Line 264 typo – substrate

Author Response

We would like to thank the reviewer for critical evaluation of this manuscript. His/her comments were very helpful in improving the overall quality of the article.

His/her comments were:

  • I would’ve liked to have seen some proof of concept biochemical data to support the in silico predictions, but probably beyond the scope of this study.

Response: Although the theoretical basis for the models used in our study has been proven by many experiments, our study is in fact only theoretical in nature. As mentioned in our manuscript, one of the reasons for performing such computational studies is that they provide important guidelines for prioritizing targets before performing costly high-throughput screening. Therefore, proof of concept would require performing high-throughput screening for multiple enzymes in the pathway, which is not feasible and beyond the scope of this article.

  • Figure 2 – SK (Shikimate Kinase) inhibitor should have a flat line and not a wedge for the carboxylic acid group (no stereochemistry here)

Response: Figure 2 has been corrected according to the reviewer's comment. However, because one of the other reviewers noted that the first three figures belong to a review article rather than a research article, all three figures were moved to the Appendix and referenced in the original manuscript.

  • Could mention AlphaFold at some point in the intro – this will no doubt serve as an important tool for drug discovery.

Response: AlphaFold has is now mentioned in the text on page 2, line 29 where it now states:

“For proteins with 3D structures that are known or predicted using AI systems such as AlphaFold, several computational methods can be used that can provide medicinal chemists with important guidelines for prioritizing targets before performing costly high-throughput screening.”

  • Figure 4 – Comparisson spelt incorrectly?

Response: Comparisson was corrected to comparison.

  • Line 264 typo – substrate

Response: Typo was corrected

Reviewer 2 Report

This study by Rok Frlan presented in silico analyses of enzymes of the shikimate pathway. The results from this study suggested that the substrate binding sites of most enzymes in this pathway are suitable targets for broad-spectrum antibacterial therapy, except for the substrate binding site of 3-de-16 oxy-D-arabino-heptulosonate-7-phosphate synthase. This study provided new insight into properties of enzymes of the shikimate pathway. Findings from this study will be helpful for the development of new antibacterial drugs targeting the shikimate pathway. However, extensive revisions are required to improve the quality of this manuscript. More efforts should be dedicated to the writing of the manuscript, especially the presentation of the major findings.

Specific comments are as follows.

Abstract

It is better to present more specific findings in the Abstract. For example, please specify “the prospects of these enzymes”, please define “ligandable or druggable”

Introduction

This section needs major revision. My suggestion is to keep main points of the background info, while leaving out well-known details. Figure 1 and Figure 2 is more suitable for a review article. It is advisable to mention these facts briefly with the literature cited. Figure 3 can be deleted or changed to a simplified scheme.

Lines 84-86. Please rephrase this sentence to be more reader friendly.

Combine the paragraph of Lines 130-136 with the preceding paragraph.

Results

This section also needs major revision. The authors should focus on major findings of this article. For example, Lines 148-172. Move some of the details to Methods. Conclusion of the first section need to be strengthened, especially for the other four enzymes of the seven enzymes. Please specify why the focus was shifted to DAHPS, DHQase, and SK.

Author Response

We would like to thank the reviewer for the critical evaluation of this manuscript. His/her comments were very helpful in improving the overall quality of the article.

Abstract

  • It is better to present more specific findings in the Abstract. For example, please specify “the prospects of these enzymes”, please define “ligandable or druggable”

Although it is difficult to present more specific results while keeping within the 250-word limit in the abstract, we changed the first half of the abstract slightly.

Additional explanation was provided as to why these targets are considered promising for antibacterial drug development. The first sentence was changed to: »Enzymes belonging to the shikimate pathway have long been considered promising targets for antibacterial drugs because they have no counterpart in mammals and are essential for bacterial growth and virulence

The druggability term was also explained: »Despite decades of research, there are currently no clinically relevant antibacterial drugs targeting any of these enzymes, and there are legitimate concerns about whether they are sufficiently drugable, i.e., whether they can be adequately modulated by potent small drug-like molecules

The term ligandability was omitted because druggability is a more important property and because the term ligandability occurs only once in the abstract. It would take too many words to explain both terms.

Next, we deleted the phrase "the prospects of these enzymes" and changed the entire sentence to: »In the present work, a combination of in silico analyses of evolutionary conservation and ligandability was performed to determine whether these enzymes are candidates for broad-spectrum antibacterial therapy.«

  • Introduction
    • This section needs major revision. My suggestion is to keep main points of the background info, while leaving out well-known details. Figure 1 and Figure 2 is more suitable for a review article. It is advisable to mention these facts briefly with the literature cited. Figure 3 can be deleted or changed to a simplified scheme.

Figures 1-3 have been moved to the supplementary and the beginning of the manuscript has been shortened.

  • Lines 84-86. Please rephrase this sentence to be more reader friendly.

The sentence was rephrased. It now reads: »Since the number of submicromolar inhibitors is insufficient to make a reliable prediction, such an analysis could not be performed with sufficient reliability for most enzymes of shikimate pathway

  • Combine the paragraph of Lines 130-136 with the preceding paragraph.

Both paragraphs were combined.

    • This section also needs major revision. The authors should focus on major findings of this article. For example, Lines 148-172. Move some of the details to Methods. Conclusion of the first section need to be strengthened, especially for the other four enzymes of the seven enzymes. Please specify why the focus was shifted to DAHPS, DHQase, and SK.

I agree that this part of the manuscript is partly too technical in lines 148-172 and has been shortened. The original text has been moved to Methods. The focus was shifted to DAHPS, DHQase, and SK because readers would not understand why DHQase I and DHQase II were analysed separately and, on the other hand, in the case of DAHPS, both isoenzymes were treated as one enzyme and in the case of SK, only one isoenzyme was used. These details are not important for other enzymes that occur in one form, without isoenzymes. However, since it is obvious that this part of the manuscript is not clear enough or might confuse the reader, I have added a sentence at the beginning of this paragraph mentioning that the other four enzymes occur only in one form.

Reviewer 3 Report

The manuscript entitled ‘Evolutionary Conservation and Druggability Analysis of Enzymes Belonging to the Bacterial Shikimate Pathway’ deals with antimicrobial target analysis with enzymes of the Shikimate pathway. The work is extensive and data are presented systematically.

  • Please mention the name of the inhibitors along with enzymes in Figure 1.
  • It is better to put biosynthetic steps of the bacterial shikimate pathway and comparison of gene clusters of bacterial, plant, and others in Figure 2.
  • Without analyzing ADMET and pharmacokinetic analysis of ligands, how would you claim druggability?
  • For better understanding to readers, please clearly mention particular ligand(s) used to dock with each enzyme in the tabulation.
  • Please mention method validation in your computational work.

Author Response

We would like to thank the reviewer for his/her critical evaluation of this manuscript. His/her comments were very helpful in improving the overall quality of the article.

  • Please mention the name of the inhibitors along with enzymes in Figure 1.

The inhibitors have been numbered from 1-9 in Figure 2. In cases where a specific name already existed in the literature, a name for each compound was also provided. However, because one of the other reviewers noted that the first three figures belonged to a review paper rather than a research article, all three figures were therefore moved to the supplementary and referenced in the original manuscript.

  • It is better to put biosynthetic steps of the bacterial shikimate pathway and comparison of gene clusters of bacterial, plant, and others in Figure 2.

I am not sure if I understood the reviewer correctly. I agree that information is given about the gene names encoding each protein in the pathway. Therefore, this information was additionally listed next to each enzyme in Figure 3 (Now Fig. S3). It is true that there are major differences between bacterial and eukaryotic enzymes. In bacteria, these enzymes are monofunctional, unlike in eukaryotes, where steps 2 to 7 are catalysed by a pentafunctional enzyme complex. However, I do not see how a comparison between different kingdoms would improve the quality of this manuscript, given the focus on bacterial enzymes. Furthermore, if you were to include such a comparison, you would have to slightly change Fig. 3, which is already filled with information and also explain it in the introductory text.

  • Without analyzing of ligands, how would you claim druggability?

If a target is druggable it can bind a small molecule with the appropriate chemical properties with the required binding affinity. Both drug-like molecules and targets must be compatible in terms of their physicochemical properties. This is also the theoretical basis for assessing the druggability of the target molecule before the small molecular drug-like ligand is known and ADMET/pharmacokinetic analysis is performed. In recent years, a variety of methods and approaches for predicting or estimating protein druggability have been reported, and the development of tools for this purpose is currently a very active area of research.

  • For better understanding to readers, please clearly mention particular ligand(s) used to dock with each enzyme in the tabulation.

I also had difficulties in understanding this comment because no docking was performed in this work. Therefore, I assumed that the reviewer meant docking of small fragments, called probes, used by FTMap and FTSite to estimate heatmaps and binding sites. The following explanation was therefore included in the Material and Methods section: “The following probes are used by each server: Acetamide,               acetonitrile, acetone, acetaldehyde, methylamine, benzaldehyde,   benzene,             isobutanol,          cyclohexane, N,N-dimethylformamide, dimethyl ether,    ethanol, ethane, phenol, isopropanol, and urea.”

Since I was not sure if the reviewer meant this change, Figure 6 (Now Fig. 3) was also slightly changed and the names of the ligands present in each binding site of each PDB structure were added.

  • Please mention method validation in your computational work.

The computational work was not validated because we only used models and techniques that have been validated by other authors.

Round 2

Reviewer 2 Report

I have no further comments.